# Health-related quality of life and associated factors among family caregivers of patients with cancer in oncologic centers of Northwest Ethiopia

Fasil Bayafers Tamene[1]*, Endalamaw Aschale Mihiretie[2], Akalu Fetene Desalew[1], Fasika Argaw Tafesse[3], Samuel Agegnew Wondm[1]

1 Department of Pharmacy, College of Medicine and Health Science, Debre Markos University, Debre Markos, Ethiopia, 2 Department of Pharmacy, College of Medicine and Health Science, Bahir Dar University, Bahir Dar, Ethiopia, 3 Department of Pharmacy, College of Medicine and Health Science, Dilla University, Dilla, Ethiopia

* fasil.baya@gmail.com

## Abstract

### Background

Providing care for individuals dealing with long-term illnesses like cancer demands significant amounts of time, energy, and emotional investment, potentially resulting in a challenging and overwhelming quality of life for those providing the care.

### Objective

The purpose of this study was to assess the level of health-related quality of life (HRQoL) and associated factors among family caregivers of patients with cancer in oncologic centers of Northwest Ethiopia.

### Method

A cross-sectional study was conducted among 412 family caregivers of patients with cancer who were following treatment at oncologic centers in Northwest Ethiopia from August to October 2023. Systematic random sampling was used to enroll study participants. Epi-data version 4.6.1 and SPSS version 26 were used for data entry and analysis, respectively. The relationship between quality of life and independent variables was examined using linear regression. Statistical significance was determined for variables having a p-value of less than 0.05 at a 95% confidence range.

### Result

A total of 412 eligible caregivers were included in the study out of 422 approached samples, yielding a 97.6% response rate. The mean score of the overall Quality of Life Brief—Scale Version was 52.7 ± 9.57. Being Spouse (β = -3.39; 95% CI: -6.49, -0.29), presence of chronic illness (β = -3.43; 95% CI: -5.56, -1.31), depression, (β = -2.55; 95% CI: -4.34, -0.75), anxiety (β = -3.27; 95% CI: -5.22, -1.32),and social support, (β = -3.61; 95% CI: -6.20, -1.02) were negatively associated with quality of life.

**Data Availability Statement:** All relevant data are within the paper and its Supporting Information files.

**Funding:** The author(s) received no specific funding for this work.

**Competing interests:** The authors have declared that no competing interests exist.

## Conclusion and recommendation

In this study, the psychological domain had the lowest mean score. Caregivers being as spouse, who were with chronic illness, manifested depression and anxiety and had a poor social support needs attention and support to improve HRQoL.

## Introduction

Cancer is regarded as the prominent ailment of the modern era, being a primary contributor to global mortality and morbidity. Despite advancements in diagnosis and treatment, its cure remains elusive. According to the World Health Organization (WHO), cancer ranks among the foremost causes of death from disease. The International Agency for Research on Cancer projected that in 2020 alone, there were 18.1 million diagnosed cases globally, with this number anticipated to rise to 27 million over the next two decades [1].

In low-income nations with restricted logistical resources, cancer is becoming a significant obstacle to maintaining citizens' quality of life [2]. Like other chronic illnesses, cancer presents various complications and can limit self-reliance, necessitating assistance from caregivers within the patient's close network, often referred to as informal or non-professional caregivers. While these caregivers play a crucial role in aiding and comforting cancer patients, they often bear a substantial burden across multiple domains, which can profoundly affect their own quality of life [3, 4].

The World Health Organization (WHO) has defined QOL as "individuals' perceptions of their position in life in the context of the culture and value systems in which they live and in relation to their goals, expectations, standards and concerns [5]." The profound impact of cancer extends beyond patients to their families [6]. Providing care for individuals with long-term illnesses such as cancer demands significant time, effort, and emotional investment, potentially resulting in a highly stressful and burdensome quality of life for caregivers [7]. Furthermore, the effect of cancer on the health-related quality of life (HRQOL) of family caregivers can hinder their capacity to effectively care for and support the patients [8]. A multitude of factors affected the QOL of family caregivers of adult cancer patients. These may include sociodemographic factors such as age, gender, place of residence, marital status, religion, educational level, occupation, monthly income, family size, types of relation to a patient and caregiving duration [9–16] and clinical social support related variables of caregivers and patients such as presence of chronic illness, depression, anxiety, level of social support, type of cancer, hospital admission of the patients, type of treatment and performance status [17–23].

In our country, caregiving for an ill family member holds deep cultural significance. Understanding the importance of health-related quality of life (HRQoL) as a health measure, it becomes essential to investigate the factors impacting caregivers' HRQoL. In economically challenged nations like Ethiopia, there is limited knowledge about the HRQoL of family caregivers of adult cancer patients. Thus, the purpose of this research was to assess the level of health-related quality of life and associated factors among family caregivers of patients with cancer in oncologic centers of Northwest Ethiopia.

## Methods

### Study setting, period and design

A cross sectional study was conducted at the University of Gondar Comprehensive Specialized Hospital (UoGCSH), Felege-Hiwot Comprehensive Specialized Hospital (FHCSH), and Tibebe-Ghion Comprehensive Specialized Hospital (TGCSH) from August to October 2023.

## Population, inclusion and exclusion criteria

All family caregivers of patients with cancer at the oncologic centers of the northwest Ethiopia were the source population. All family caregivers of cancer patients at UoGCSH, FHCSH and TGCSH who fulfills the inclusion criteria during the data collection period were study population. Caregivers with the age of 18 years and above, caring for patients with definitive diagnosis of cancer and those who can be identified by the patient as being caregivers was included in the study. Caregivers with psychiatric conditions and those who were unwilling to participate were excluded from the study.

## Sample size determination

Single population proportion formula was used to determine the number of participants needed for this study. To get better representative sample, we used proportion as 50% in sample size calculation. $n = \frac{(Z\alpha/2)^2 \times p(1-p)}{d^2}$, where n is the desired sample size for a population of >10,000, Z is the typical normal distribution set at 1.96 (which corresponds to 95% CI), the p-value signifies that positive prevalence was utilized in calculating the optimal sample size, and W is the degree of accuracy 0.05 required (a marginal error is 0.05). Then computing for n = $1.96^2{}^*0.5(1–0.5)/0.05^2$, n = 384. By adding 10% non-response rate, the final calculated sample size was 422.

## Sampling technique and procedure

The total number of cancer patients over a three-month period was extracted from patient registration documents to allocate respondents proportionally across study areas. Following proportional allocation, a systematic random sampling method was employed to choose study participants. The sampling fraction (k) was determined by dividing the total number of cancer patients in each study area by the overall sample size; (925/422 equals 2.19, approximated to 2). Proportional allocation was then calculated for each study area (UOGCSH: 422x358/925 = 163, FHCSH: 422x376/925 = 172, TGCSH: 422*191/925 = 87). A starting point was randomly selected from 1 or 2. Subsequently, participants were interviewed, and relevant data was simultaneously gathered from medical charts for every two patients until the required sample size was achieved. A unique patient identification card number was assigned as a questionnaire code to prevent duplicate inclusion of the same patient in the study.

## Study variables

Health-related quality of life was the main outcome variable. The independent variables were the sociodemographic characteristics of the participants (like sex, age, marital status, educational status, occupation, monthly income, patient's health insurance status, relationship to patient, family size, duration of caregiving and support from other family members) and clinical related variables (i.e. presence of chronic disease of caregiver, presence of substance use, depression, anxiety, and eastern cooperative oncology group performance status) and level of social support.

## Operational definitions

**Anxiety.** Individuals who score ≥10 from GAD-7 were considered as having anxiety [24].
**Depression.** Individuals who are found to score≥10 from PHQ-9 were considered as having depression [25].

**Health related quality of life.** Based on the WHOQOL-BREF 26 items, an individuals with the score approaches to 100 indicates the best possible quality of life, and individuals with the score approaches to zero indicates the poorest quality of life [26].

**Social support.** According to the Oslo-3 social support scale which ranges from 3 to 14, individuals who score between 3 and 8 are considered to have poor social support, who score of 9–11 is considered as having moderate social support, and score of 12–14 is considered as having strong social support [27].

**Substance use.** Current users: using at least one of a specific substance (alcohol, Khat or cigarettes) for nonmedical purposes within the last 3 months, according to the alcohol, smoking, and substance involvement screening tool (ASSIST) [28].

## Data collection instrument, procedures, and quality control

A structured questionnaire, based on existing literature [29], underwent adaptations to suit the study area's context and the socio-demographic characteristics of the participants. This questionnaire was translated into the local Amharic language and subsequently back-translated into English to ensure consistency. Variables obtained from patients' medical records did not necessitate translation. The data collected by interviewing family caregivers includes sociodemographic characteristics, presence of chronic disease, substance use, anxiety, depression and social support. The patients' medical charts were used to fill in clinically related variables like cancer diagnosis of the patient, duration of cancer diagnosis and treatment, current treatment modality, functional status and metastasis.

The data collection tool has seven parts. The first part contains socio-demographic characteristics of the study participants like age, sex, marital status, monthly income, residence, religion, educational status, occupation, patient's health insurance status, relationship to patient, duration of caregiving, care support from other family members, duration of caregiving per month and number of family size. The second section consisted of clinical related variables like presence of chronic disease, presence of substance use, depression and anxiety of caregiver, cancer diagnosis of the patient, duration of cancer diagnosis and treatment, current treatment modality, ECOG(Eastern Cooperative Oncology Group), and presence of metastasis.

The third section consisted HRQoL measuring tool. Health related quality of life was assessed by utilizing the World Health Organization Quality of Life Scale–Brief version (WHOQOL–BREF) which is a 26-item self-administered generic questionnaire. The WHO-QOL-BREF is a sound, cross-culturally valid assessment of QoL, as indicated by its four domains: physical, psychological, social, and environment [30]. It creates a profile with four domain scores: physical health (7 items), psychological health (6 items), social relationships (3 items), and environmental domain (8 items) as well as two individually scored items concerning the individuals' impression of their quality of life (QI) and health (Q2). The scores of items in each domain will be combined, and the mean or median will be calculated to produce the domain score [31]. WHOQOL–BREF has been used in Ethiopia [32].

The fourth section of the questionnaire is assessing social support by the Oslo-3 social support scale. The OSS-3 scores ranged from 3–14, with a score of 3–8 = poor social support; 9–11 = moderate social support; and 12–14 = strong social support [27]. The fifth section involved the current substance use assessment tool. ASSIST was utilized to screen participants briefly for the use of psychoactive substances. This tool was developed and validated by the WHO [28].

The sixth section consisted assessment of depression by using the (**PHQ-9**). The PHQ-9 score ranges from 0 to 27. Each of the 9 items was scored from 0 (not at all) to 3 (nearly every day). A PHQ-9 score 0–4 indicates minimal/no depression, 5–9 indicates mild depression, 10–14 indicates moderate depression, 15–19 indicates moderately severe depression and a score of

20 to 27 indicates severe type of depression [33]. Moreover, PHQ-9 has been validated in Ethiopian healthcare context with specificity and sensitivity of 67% and 86%, respectively. A cut-off point of 10 or more has been used to screen for depression [25].

The seventh section of the questionnaire assessed anxiety by using **GAD-7**. Scores of 0, 1, 2, and 3 were assigned to the response categories of "not at all," "several days," "more than half the days," and "nearly every day," respectively, and the scores for the seven questions were added together. Using the threshold score of 10, the GAD-7 has a sensitivity of 89% and a specificity of 82% for GAD [24].

Data collection involved face-to-face interviews utilizing a pretested and structured questionnaire. At UoGCSH and FHCSH, three nurses with bachelor's degrees, supervised by a senior oncology nurse holding a master's degree, conducted the interviews. Similarly, at TGCSH, two nurses with bachelor's degrees, overseen by a senior oncology nurse with a bachelor's degree, conducted the interviews. Each day, supervisors provided necessary supplies to the data collectors and ensured the accuracy of completed questionnaires, addressing any issues promptly. The lead researcher provided resources for all study areas.

## Data entry and analysis

The collected data underwent cleaning, coding, and entry into Epi Data 4.6.0, followed by analysis using Statistical Package for Social Studies (SPSS) version 26. Descriptive analysis utilized measures such as mean with standard deviation (SD), frequency, and percentages to assess data distribution. Bivariate and multivariate linear regression analyses were conducted to explore the relationship between HRQoL and independent variables.

Linearity was confirmed via scatter plots, demonstrating a negative association between HRQoL and all continuous independent variables. Normality assumption was verified through histograms and normal probability-probability (P-P) plots, indicating a normal distribution with skewness ranging between ±1, signifying normality. The Durbin-Watson statistic ranged from 1.5 to 2.5, affirming independent observations. Multicollinearity was assessed, with the maximum Variance Inflation Factor (VIF) found to be less than 5, indicating acceptability.

Variables with a p-value < 0.25 in simple linear regression were chosen for multiple linear regressions. A p-value < 0.05 was considered indicative of an independently associated factor in multivariable linear regression. Model fitness was evaluated, with statistical significance observed at F = 5.13, P value < 0.001, R = 0.681, $R^2$ = 0.431, and adjusted $R^2$ = 0.357. Unstandardized beta coefficients with a 95% confidence interval (CI) were utilized to assess the level of association and statistical significance in multiple linear regression analysis

## Ethics approval and consent to participate

The study obtained ethical approval from the institutional review board of the College of Medicine and Health Sciences at Bahir Dar University, with reference number 796/2023. Prior to their participation, all study participants were fully informed about the study's objectives and provided written consent. Measures were taken to ensure participants' privacy, and personal identifiers were omitted from the data collected. The study adhered to the principles outlined in the Declaration of Helsinki.

## Results

### Sociodemographic characteristics among family caregivers of cancer patients

Out of a total of 422 individuals approached, 412 eligible caregivers were enrolled in the study, resulting in a response rate of 97.6%. The majority (51.9%) were female, with an average age of

36.1 (±12.2) years. Nearly half (48.3%) of the participants were married, and more than two-thirds (72.6%) resided in urban areas. About a third (31.6%) of respondents had completed secondary school education, and a similar proportion (34.7%) were employed in the private sector. Close to half (45.9%) of participants reported a monthly income of less than 3000 ET birr, while over two-thirds (67.2%) had health insurance coverage. Regarding their relationship to the patient, more than a third (34.7%) were spouses, and over half (55.3%) had families consisting of 4 to 7 members. Approximately two-thirds (64.3%) of participants had been providing care for a duration ranging from 12 to 36 months, and nearly all (90.5%) received support from other family members (Table 1).

## General characteristics among cancer patients

The majority of patients (60.2%) were female, with an average age of 41.91 (±16.5) years. Around a quarter (24.0%) had been diagnosed with breast cancer, and most of them (61.7%) had been living with the illness for less than a year. Over half (58.5%) had undergone treatment for less than a year, with the majority (63.3%) receiving chemotherapy. A quarter (25.0%) had an ECOG score of III, and more than a third (36.7%) had cancer that had metastasized. About one-seventh (14.6%) reported a family history of cancer (Table 2).

## Clinical, substance and psychological related characteristics among family caregivers of cancer patients

In relation to clinical variables, about a quarter (25.2%) of participants had chronic illnesses, with diabetes mellitus being the most common. Nearly one-third (31.1%) reported using psychoactive substances, primarily alcohol. A significant portion (42.5%) experienced symptoms of depression, and close to one-third (30.6%) reported experiencing anxiety. Additionally, over half (55.1%) of participants reported poor social support (Table 3).

## Self-rated perceived quality of life and health satisfaction among family caregivers of cancer patients

More over one-third(38.6%) of participants said they had a low HRQoL. About one-third (33.7%) of respondents indicated dissatisfaction with their level of health satisfaction, while slightly more than a quarter (29.0%) expressed neither satisfied nor dissatisfied (Table 4).

## Health related quality of life among family caregivers of cancer patients

The study found that family caregivers of cancer patients had an average quality of life score of 52.7 (95% CI: 51.70, 53.63), with a standard deviation of 9.57. The range of the participants' overall quality of life scores was 26.5 to 81.5. The psychological domain had the lowest average score (47.82±16.61) among the four quality of life domains, whereas the environmental domain had the highest mean score (58.30±17.73) (Table 5).

## Factors associated with health related quality of life among family caregivers of cancer patients

After adjusting for confounders by applying multiple linear regression analysis, a significant association was found between HRQoL and some predictor variables. Thus, participants who had a spouse had an overall HRQoL that was 3.39 times poorer than those who had a brother or sister relationship with the patient (B = -3.39; 95% CI: -6.49, -0.29). When compared to respondents without a chronic illness, those with a chronic illness had an overall HRQoL that was 3.43 times poorer (β = -3.43; 95% CI: -5.56, -1.31). When it came to feeling depression,

**Table 1. Sociodemographic characteristics among family caregivers of cancer patients (n = 412).**

| Variables | Categories | Frequency | Percentage |
|---|---|---|---|
| Sex | Male | 198 | 48.1 |
| | Female | 214 | 51.9 |
| Age | 18–28 | 77 | 18.7 |
| | 29–39 | 162 | 39.3 |
| | 40–50 | 104 | 25.2 |
| | 51–60 | 58 | 14.1 |
| | >60 | 11 | 2.7 |
| Marital status | Single | 168 | 40.8 |
| | Married | 199 | 48.3 |
| | Divorced | 26 | 6.3 |
| | Widowed | 19 | 4.6 |
| Residence | Urban | 299 | 72.6 |
| | Rural | 113 | 27.4 |
| Religion | Orthodox | 330 | 80.1 |
| | Muslim | 70 | 17.0 |
| | Protestant/Catholic | 12 | 2.9 |
| Educational level | No formal education | 26 | 6.3 |
| | Primary (1–8 grades) | 75 | 18.2 |
| | Secondary (9–12 grade) | 130 | 31.6 |
| | Diploma and above | 181 | 43.9 |
| Occupation | Farmer | 29 | 7.0 |
| | Government employee | 79 | 19.2 |
| | Private employee | 143 | 34.7 |
| | Student | 36 | 8.7 |
| | Jobless | 73 | 17.7 |
| | Housewife | 52 | 12.6 |
| Monthly income in birr | <3000 ET. birr | 189 | 45.9 |
| | 3001–7000 ET. birr | 166 | 40.3 |
| | 7001–10000 ET birr | 33 | 8.0 |
| | >10000 ET. birr | 24 | 5.8 |
| Health insurance | Yes | 277 | 67.2 |
| | No | 135 | 32.8 |
| Relationship to patient | Spouse | 143 | 34.7 |
| | Parent | 54 | 13.1 |
| | Son/daughter | 135 | 32.8 |
| | Brother/sister | 80 | 19.4 |
| Family size | ≤3 | 39 | 9.5 |
| | 4–7 | 228 | 55.3 |
| | ≥8 | 145 | 35.2 |
| Duration of care giving (per month) | < 12 months | 66 | 16.0 |
| | 12–36 Months | 265 | 64.3 |
| | > 36 months | 81 | 19.7 |
| Care support from other family member | Yes | 373 | 90.5 |
| | No | 39 | 9.5 |
| Family history of cancer | Yes | 60 | 14.6 |
| | No | 352 | 85.4 |

1 ET. birr = 0.017 USD

**Table 2. General characteristics of cancer patients (n = 412).**

| Variables | Categories | Frequency | Percent |
|---|---|---|---|
| Sex | Male | 164 | 39.8 |
| | Female | 248 | 60.2 |
| Age | Mean (SD) = 41.91 (±16.5) | | |
| Cancer type | Breast | 99 | 24.0 |
| | Cervical | 86 | 20.9 |
| | Colorectal | 53 | 12.9 |
| | Lymphoma | 34 | 8.3 |
| | Ovarian | 32 | 7.8 |
| | Lung | 30 | 7.3 |
| | Bone and soft tissue | 28 | 6.8 |
| | Head and neck | 21 | 5.1 |
| | other* | 29 | 7.0 |
| Duration of illness | < 1 year | 254 | 61.7 |
| | 1–5 year | 141 | 34.2 |
| | >5 year | 17 | 4.1 |
| Duration of Treatment | < 1 year | 241 | 58.5 |
| | 1–5 year | 162 | 39.3 |
| | >5 year | 9 | 2.2 |
| Type of treatment | Chemotherapy | 261 | 63.3 |
| | Surgery | 39 | 9.5 |
| | Chemotherapy and surgery | 112 | 27.2 |
| Functional status (ECOG) | I | 152 | 36.9 |
| | II | 138 | 33.5 |
| | III | 103 | 25.0 |
| | IV | 19 | 4.6 |
| Metastasis | Yes | 151 | 36.7 |
| | No | 261 | 63.3 |

*, Hodgkin's lymphoma, Pancreatic cancer, Squamous cell carcinoma, Multiple myeloma, Hepato-cellular carcinoma

individuals who showed symptoms had an overall HRQoL that was 2.55 times lower than those who did not ($\beta$ = -2.55; 95% CI: -4.34, -0.75). Likewise, individuals experiencing anxiety had an overall HRQoL that was 3.27 times lower than that of patients who did not experience anxiety ($\beta$ = -3.27; 95% CI: -5.22, -1.32). In relation to social support, caregivers with inadequate social support experienced a 3.61-fold reduction in total HRQoL compared to those with strong social support ($\beta$ = -3.61; 95% CI: -6.20, -1.02 (Table 6).

## Discussion

Cancer inflicts extensive physical, emotional, and social hardships on individuals and their families. Family caregivers navigate through an adjustment and acceptance phase in dealing with the illness, managing multiple responsibilities under significant stress levels, leading to physical, emotional, social, and financial strain, ultimately impacting their quality of life [1]. In an effort to evaluate the level of HRQoL and its determinants among family caregivers of cancer patients, this study revealed that the mean overall HRQoL score among participants was 52.7, with a standard deviation of 9.57. Among the four domains, the psychological domain exhibited the lowest mean score (47.82±16.61). Furthermore, the study identified predictor variables influencing HRQoL among the participants. Thus Being Spouse ($\beta$ = -3.39; 95% CI:

**Table 3. Clinical, substance and psychological related characteristics among caregivers of cancer patients (n = 412).**

| Variables | Categories | Frequency | Percent |
|---|---|---|---|
| Presence of chronic illness* | yes | 104 | 25.2 |
| | No | 308 | 74.8 |
| Substance use | Yes | 128 | 31.1 |
| | No | 284 | 68.9 |
| Type of substance use | Alcohol | 89 | 69.6 |
| | Khat | 26 | 20.3 |
| | Cigarette | 13 | 10.1 |
| Depression | Yes | 175 | 42.5 |
| | No | 237 | 57.5 |
| Anxiety | Yes | 126 | 30.6 |
| | No | 286 | 69.4 |
| Social support | Poor | 227 | 55.1 |
| | Moderate | 124 | 30.1 |
| | Strong | 61 | 14.8 |

*Hypertension, DM, Cardiac disorder, Respiratory disorder, Arthritis, PUD, HIV

-6.49, -0.29), presence of chronic illness (β = -3.43; 95% CI: -5.56, -1.31), depression, (β = -2.55; 95% CI: -4.34, -0.75), anxiety (β = -3.27; 95% CI: -5.22, -1.32), and social support, (β = -3.61; 95% CI: -6.20, -1.02) were negatively associated with HRQoL. Psychological domain was the lowest which is in line with various evidences [18, 34–36]. In contrast, findings from Turkey [37, 38]. This difference could potentially be attributed to socio-cultural factors, variations in sample sizes, and the utilization of validated tools to measure HRQoL.

Participants who were spouses of the patients exhibited lower overall HRQoL compared to those who were siblings. This observation aligns with previous research conducted in Ethiopia [29] and China [12, 39]. It is expected that the daily lives of cancer patients and their caregivers will be significantly disrupted. Furthermore, family caregivers are typically involved in all aspects of cancer care and management, requiring them to invest time, financial resources, and physical effort in navigating the complexities of cancer treatment. This caregiving burden can negatively affect their physical and mental well-being. Thus, caring for a spouse with cancer can be overwhelming, leading to increased psychological stress and a diminished quality of life [39]. Additionally, spouses often bear the brunt of caregiving responsibilities and are less likely to receive external support, which can further impact their quality of life [29].

**Table 4. Self-rated perceived quality of life and health satisfaction among family caregivers of cancer patients (n = 412).**

| Variable | Category | Frequency (%) |
|---|---|---|
| Perceived quality of life | Very poor | 44 (10.7) |
| | Poor | 159 (38.6) |
| | Neither poor nor good | 141 (34.2) |
| | Good | 48 (11.7) |
| | Very good | 20 (4.9) |
| Perceived health satisfaction | Very dissatisfied | 74(18.0) |
| | dissatisfied | 139 (33.7) |
| | Neither satisfied nor dissatisfied | 120 (29.1) |
| | Satisfied | 42 (10.2) |
| | Very satisfied | 37 (9.0) |

**Table 5. Health related quality of life among caregivers of cancer patients (n = 412).**

| Domains | Mean ± SD | 95% CI | Percent of participants who scored below the mean |
|---|---|---|---|
| Physical | 53.25±21.72 | (50.94, 55.39) | 54.4 |
| Psychological | 47.82±16.61 | (46.18, 49.43) | 63.1 |
| Social relationship | 51.44±19.48 | (49.64, 53.48) | 54.9 |
| Environmental | 58.30±17.73 | (56.51, 60.14) | 42.2 |
| Overall HRQoL | 52.70±9.57 | (51.70, 53.63) | 40.4 |

Regarding the existence of chronic illness, individuals with chronic conditions experienced a decrease in overall HRQoL compared to those without such conditions [21, 37, 40, 41]. This phenomenon can be explained by the fact that physical well-being is a crucial aspect of quality of life for caregivers of oncology patients, given the physically and emotionally demanding nature of caregiving. As a result, healthcare professionals should take action by integrating caregivers into comprehensive cancer care protocols, encouraging them to prioritize their own health issues, promoting self-care practices, and offering guidance on accessing community and healthcare services [42].

Regarding the experience of depression and anxiety, individuals who exhibited these conditions had diminished overall HRQoL compared to those who did not. This finding aligns with several previous studies [22, 40, 43–46]. The reason behind this correlation may lie in the significant physical, emotional, and practical ramifications that a cancer diagnosis has on the lives of both patients and their families. The responsibilities placed on family caregivers of cancer patients are known to heighten caregiver strain, negatively impact mental and physical health, and worsen their quality of life [47]. Therefore, caregivers experiencing depression and anxiety should undergo proper screening and management to enhance patient care.

In terms of social support, caregivers who reported inadequate social support exhibited lower overall HRQoL compared to those who had strong social support. This observation is consistent with various studies [20, 48, 49]. The absence of adequate social support can signal both the level of direct care provided by caregivers to patients and their opportunities for relief from caregiving responsibilities. Therefore, enhancing both psychosocial support and available resources, such as home care services, could potentially enhance the quality of life for these caregivers.

## Strength and limitation of study

This study, which identified the level of health-related quality of life and its predictors, is the first of its kind in Northwest Ethiopia. Furthermore, in a multi-facility environment, employing a comparatively larger sample size and focusing on the most likely prospective determinant variables, which may be a component of generalization. Nevertheless, the study's self-reported results depend on the veracity of the subjects and could be subjected to recall bias. Assessment of substance usage is also vulnerable to social desirability bias. The study's cross-sectional design made it impossible to demonstrate a causal association and precluded a follow-up. The WHO BREF assessment tool is not validated in the Ethiopian setting, even if it is cross-culturally valid.

## Conclusion and recommendation

This study found that psychological domain had the lowest mean score compared with other HRQoL measuring domains suggesting a potential need for enhanced psychological support. Furthermore, there is a crucial need to prioritize efforts to bolster social support, particularly

**Table 6. Multivariate linear regression for overall quality of life and associated factors among family caregivers of cancer patients (n = 412).**

| Variables | Categories | SLR: β (95%CI) | MLR: β (95%CI) | P-value |
|---|---|---|---|---|
| Sex | Male | -1.32 (-3.17, 0.52) | 0.52 (-1.80, 2.84) | 0.661 |
| | Female | 1 | 1 | |
| Age | >60 | 5.54 (-0.51, 11.59) | 3.70 (-2.21, 9.62) | 0.219 |
| | 51–60 | -0.87 (-4.14, 2.38) | -0.81 (-4.08, 2.46) | 0.627 |
| | 40–50 | 1.50 (-1.31, 4.32) | 0.70 (-2.11, 3.51) | 0.625 |
| | 29–39 | 1.17 (-1.42, 3.76) | 0.31 (-2.21, 2.83) | 0.807 |
| | 18–28 | 1 | 1 | |
| Marital status | Single | 0.20 (-1.77, 2.17) | 2.12 (-.28, 4.53) | 0.084 |
| | Divorced | 0.13 (3.79, 4.05) | 1.55 (-2.30, 5.42) | 0.428 |
| | Widowed | -3.59 (-8.00, 0.82) | -3.67 (-7.95, 0.60) | 0.092 |
| | Married | 1 | 1 | |
| Educational level | No formal education | 2.54 (-1.38, 6.48) | 3.07 (-0.68, 6.84) | 0.109 |
| | Primary (1–8 grades) | 0.33 (-2.33, 3.01) | 1.21 (-1.35, 3.79) | 0.353 |
| | Secondary (9–12 grade) | 0.99 (-3.13, 1.14)) | 0.02 (-2.02, 2.07) | 0.982 |
| | Diploma and above | 1 | 1 | |
| Occupation | Farmer | 1.01 (-3.06, 5.08) | -0.62 (-4.55, 3.29) | 0.753 |
| | Private employee | 0.78 (-1.82, 3.39) | -0.06 (-2.57, 2.45) | 0.960 |
| | Student | -1.01 (-4.77, 2.76) | -0.73 (-4.36, 2.89) | 0.691 |
| | Jobless | -1.89 (-4.94, 1.14) | -0.79 (-3.84, 2.25) | 0.610 |
| | Housewife | -0.41 (-3.73, 2.91) | -2.14 (-5.39,1.10) | 0.196 |
| | Government employee | 1 | 1 | |
| Monthly income in birr | ≤ 3000 ET. birr | -2.59 (-6.46, 1.26) | -1.97 (-5.85,1.91) | 0.319 |
| | 3001–7000 ET. birr | 1.02 (-2.87, 4.92) | 1.148 (-2.77, 5.07) | 0.565 |
| | 7001–10000 ET birr | 0.61 (-4.25, 5.48) | 2.37 (-2.43, 7.17) | 0.333 |
| | >10000 ET. birr | 1 | 1 | |
| Health insurance | No | -1.31 (-3.28, 0.66) | -0.62 (-2.62, 1.37) | 0.540 |
| | Yes | 1 | 1 | |
| Relationship to patient | Spouse | -2.92 (-5.53, -0.41 | -3.39 (-6.49, -0.29)* | **0.032** |
| | Parent | 2.22 (-0.99, 5.44) | 0.59 (-2.55, 3.73) | 0.711 |
| | Son/daughter | 0.37 (-2.23, 2.98) | -0.22 (-2.79, 2.34) | 0.866 |
| | Brother/sister | 1 | 1 | |
| Family size | ≤3 | 0.05 (-3.32, 3.34) | -0.15 (3.46, 3.15) | 0.927 |
| | 4–7 | 2.43 (0.44, 4.42) | 1.67 (-0.30, 3.64) | 0.097 |
| | ≥8 | 1 | 1 | |
| Duration of care giving (per month) | > 36 months | -3.36 (-6.44, -0.27) | -2.36 (-5.37, 0.64) | 0.123 |
| | 12–36 months | 0.51 (-2.05, 3.07) | -0.63 (-3.23, 1.97) | 0.633 |
| | < 12 months | | 1 | |
| Care support from other family member | No | -2.55 (-5.72, 0.60) | -2.69 (-5.82, 0.43) | 0.091 |
| | Yes | 1 | 1 | |
| Presence of chronic illness | Yes | -5.17 (-7.24, -3.09) | -3.43 (-5.56, -1.31)* | **0.002** |
| | No | 1 | 1 | |
| Substance use | Yes | -1.25 (-3.25, 0.75) | -0.56 (-2.53, 1.41) | 0.577 |
| | No | 1 | 1 | |
| Depression | Yes | -2.95 (-4.80, -1.09) | -2.55 (-4.34, -0.75)* | **0.005** |
| | No | 1 | 1 | |
| Anxiety | Yes | -4.24 (-6.21, -2.27) | -3.27 (-5.22, -1.32)* | **0.001** |
| | No | 1 | 1 | |

(*Continued*)

**Table 6.** (Continued)

| Variables | Categories | SLR: β (95%CI) | MLR: β (95%CI) | P-value |
|---|---|---|---|---|
| Social support | Poor | -3.49 (-6.18, -0.81) | -3.61 (-6.20, -1.02)* | **0.006** |
| | Moderate | -0.54 (-3.45, 2.36) | -1.35 (-4.12, 1.42) | 0.338 |
| | Strong | 1 | | |
| ECOG | IV | -0.87 (-5.45, 3.71) | -0.87 (-5.19, 3.44) | 0.691 |
| | III | -1.43 (-3.38, 0.97) | -1.28 (-3.59, 1.03) | 0.276 |
| | II | -1.02 (-3.23, 1.19) | -0.86 (-2.95, 1.21) | 0.414 |
| | I | 1 | 1 | |

1 ET. birr = 0.017 USD, R = 0.681, $R^2$ = 0.431, and adjusted $R^2$ = 0.357

*$P<0.05$; ß, unstandardized beta coefficient; MLR, multiple linear regression; SLR, simple linear regression; **bold figures**; statistically significant variables

among caregivers in spousal relationships. Adequate clinical screening and follow-up are essential for individuals experiencing depression and anxiety to effectively manage these manifestations. Future research endeavors could delve deeper into exploring the causal relationship between health-related quality of life and various potential determinant variables.

## Supporting information

**S1 Checklist. Plos One human subjects research checklist.**
(DOCX)

**S2 Checklist. STROBE checklist.**
(DOCX)

**S1 File. Dataset.**
(XLSX)

**S2 File. Dataset.**
(XLSX)

## Author Contributions

**Conceptualization:** Fasil Bayafers Tamene.

**Data curation:** Fasil Bayafers Tamene.

**Formal analysis:** Fasil Bayafers Tamene, Fasika Argaw Tafesse.

**Investigation:** Fasil Bayafers Tamene, Samuel Agegnew Wondm.

**Methodology:** Fasil Bayafers Tamene, Endalamaw Aschale Mihiretie, Samuel Agegnew Wondm.

**Project administration:** Fasil Bayafers Tamene, Endalamaw Aschale Mihiretie.

**Resources:** Fasil Bayafers Tamene, Akalu Fetene Desalew, Fasika Argaw Tafesse.

**Software:** Fasil Bayafers Tamene, Samuel Agegnew Wondm.

**Supervision:** Endalamaw Aschale Mihiretie, Samuel Agegnew Wondm.

**Validation:** Fasil Bayafers Tamene, Akalu Fetene Desalew, Samuel Agegnew Wondm.

**Visualization:** Samuel Agegnew Wondm.

**Writing – original draft:** Fasil Bayafers Tamene.

**Writing – review & editing:** Fasil Bayafers Tamene, Endalamaw Aschale Mihiretie, Akalu Fetene Desalew, Fasika Argaw Tafesse, Samuel Agegnew Wondm.

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
