## [Decision Letter · Decision Letter 0]

23 Apr 2024

PONE-D-24-10340Health-related quality of life and associated factors among family caregivers of patients with cancer in oncologic centers of Northwest Ethiopia, a Multicenter cross-sectional study, 2023PLOS ONE

Dear Dr. Tamene,

Thank you for submitting your manuscript to PLOS ONE. After careful consideration, we feel that it has merit but does not fully meet PLOS ONE’s publication criteria as it currently stands. Therefore, we invite you to submit a revised version of the manuscript that addresses the points raised during the review process.

We look forward to receiving your revised manuscript.

Kind regards,

Noorsuzana Mohd Shariff

Academic Editor

PLOS ONE

Journal Requirements:

- https://doi.org/10.1371/journal.pone.0281742

- doi:10.1136/bmjopen-2023-074112

In your revision ensure you cite all your sources (including your own works), and quote or rephrase any duplicated text outside the methods section. Further consideration is dependent on these concerns being addressed.

Additional Editor Comments:

Dear author,

Thank you for considering PLOS One as the journal to publish your valuable scientific work. Your article has passed through the peer review process and both reviewers had agreed that the article has an important merit for publication and contribution to the field. However, minor revision is required. Please address all the comments and resubmit your revision.

Thank you.

Reviewers' comments:

Reviewer's Responses to Questions

**Comments to the Author**

1. Is the manuscript technically sound, and do the data support the conclusions?

Reviewer #1: Yes

Reviewer #2: Yes

2. Has the statistical analysis been performed appropriately and rigorously? 

Reviewer #1: Yes

Reviewer #2: I Don't Know

3. Have the authors made all data underlying the findings in their manuscript fully available?

Reviewer #1: Yes

Reviewer #2: Yes

4. Is the manuscript presented in an intelligible fashion and written in standard English?

Reviewer #1: Yes

Reviewer #2: Yes

5. Review Comments to the Author

Reviewer #1: Dear Authors,

Generally, the study was very good. The writing is detailed except for minor issues as following:

1) Title: please remove "2023"

2) Results: For Table 1 and 6, please put currency reference (e.g I ET birr= x USD)

Table 2: please correct the spelling for Hodgkin's lymphoma,

Table 2: please do not use * for indicating the mean, please write Mean (SD)= the actual value

Under the following sub topic (Factors associated with health related quality of life among family

caregivers of cancer patients), please rephrase the first sentence "After adjusting for multiple linear regression analysis" to After adjusting for confounders by applying multiple linear regression analysis, .....

3) Conclusion and recommendation: Please summarise your major findings first: the level of HRQoL, folowed with the associated factors before suggesting with screening etc.

Reviewer #2: Please correct some typing errors. Please see my comment in the manuscript. You have written a clear and concise manuscript. If possible, please add some more discussion on the general characteristics of cancer patients.

6. PLOS authors have the option to publish the peer review history of their article (what does this mean?). If published, this will include your full peer review and any attached files.

Reviewer #1: No

Reviewer #2: **Yes: **ROHAYU HAMI

---

## [Author Response · Author response to Decision Letter 0]

10 May 2024

Responses to the review’s comments

Dear PLOS ONE editorial team,

Thank you for giving us the opportunity to submit a revised draft of the manuscript, and we would also like to thank you for your crucial comments on our paper (Manuscript ID: PONE-D-24-10340). We are very concerned and have combined all the suggested comments provided, which we believe strengthen our paper, and we hope this will render our paper eligible for consideration for publication in your reputed journal. We appreciate the time and effort that you and the reviewers dedicated to providing feedback on our manuscript and are grateful for the insightful comments and valuable improvements to our paper.

The authors would like to inform you that we have addressed the comments and recommendations made by both reviewers and the editor, point by point. In addition, throughout our revision, we made our best corrections too. All changes in the revised manuscript are highlighted using tracking changes within the manuscript. Please see below, in blue, for a point-by-point response to the reviewers’ comments and concerns. All page numbers refer to the revised manuscript file with tracked changes.

Comments from the editor:

Journal Requirements:

#1…...Please ensure that your manuscript meets PLOS ONE's style requirements, including those for file naming. 

Author response: Thank you for your recommendationstoadhere PLOS ONE’s style of requirements. We have ensured that the manuscript, its name, figures, and tables are per the journal requirements.

#2….. We noticed you have some minor occurrence of overlapping text with the following previous publication(s), which needs to be addressed:

- https://doi.org/10.1371/journal.pone.0281742

- doi:10.1136/bmjopen-2023-074112

In your revision ensure you cite all your sources (including your own works), and quote or rephrase any duplicated text outside the methods section. Further consideration is dependent on these concerns being addressed.

Author response: 

#3…... Please provide a complete Data Availability Statement in the submission form, ensuring you include all necessary access information or a reason for why you are unable to make your data freely accessible. If your research concerns only data provided within your submission, please write "All data are in the manuscript and/or supporting information files" as your Data Availability Statement.

Author response: Thank you very much for your request to revise the data availability statement. Based on your recommendation, we have revised it and stated in the main document as well as in the cover letter as “All necessary files are available in the manuscript including the datasets as supporting files (S1 File. Dataset and S2 File. Dataset)”.

#4…..Please include captions for your Supporting Information files at the end of your manuscript, and update any in-text citations to match accordingly. Please see our Supporting Information guidelines for more information: http://journals.plos.org/plosone/s/supporting-information. 

Author response: Thank you very much for the comment and we have added the caption for all supporting information.

#5……Please review your reference list to ensure that it is complete and correct. If you have cited papers that have been retracted, please include the rationale for doing so in the manuscript text, or remove these references and replace them with relevant current references. Any changes to the reference list should be mentioned in the rebuttal letter that accompanies your revised manuscript. If you need to cite a retracted article, indicate the article’s retracted status in the References list and also include a citation and full reference for the retraction notice.

Author response: Thank you very much for your suggestion and we would like to assure you that retracted reference was not used in the manuscript. 

Response to Reviewers’ comments

Reviewer 1

#1…...Title: please remove "2023"

Author response: Thank you for your comment and we have taken it. 

#2…... Results: For Table 1 and 6, please put currency reference (e.g I ET birr= x USD)

Author response: Thank you very much for your recommendation and we have included the equivalent currency in USD.

#3…..Table 2: please correct the spelling for Hodgkin's lymphoma

Author response: Thank you very much for identifying the typing error and it is corrected. 

#4…...Table 2: please do not use * for indicating the mean, please write Mean (SD) = the actual value

Author response Thank you very much for your suggestion and correction was made. 

#5…...Under the following sub topic (Factors associated with health related quality of life among family caregivers of cancer patients), please rephrase the first sentence "After adjusting for multiple linear regression analysis" to After adjusting for confounders by applying multiple linear regression analysis,

Author response: Thank you very much for recommending the way of writing and amendment was made. 

#6…...Conclusion and recommendation: Please summarise your major findings first: the level of HRQoL, folowed with the associated factors before suggesting with screening etc.

Author response Thank you very much for your insight and we have revised the conclusion and recommendation section to make it clear. 

Reviewer 2:

#1…... Comma here 

Author response: Thank you very much for your comment and we have added the comma.

#2…... What is this number for?

Author response: Thank you very much for identifying the error made and it is corrected.

---

## [Editor Report · Decision Letter 1]

13 May 2024

Health-related quality of life and associated factors among family caregivers of patients with cancer in oncologic centers of Northwest Ethiopia

PONE-D-24-10340R1

Dear Dr. Tamene,

We’re pleased to inform you that your manuscript has been judged scientifically suitable for publication and will be formally accepted for publication once it meets all outstanding technical requirements.

Kind regards,

Noorsuzana Mohd Shariff

Academic Editor

PLOS ONE
---

## [Editor Report · Acceptance letter]

22 May 2024

PONE-D-24-10340R1 

PLOS ONE

Dear Dr. Tamene, 

I'm pleased to inform you that your manuscript has been deemed suitable for publication in PLOS ONE. Congratulations! Your manuscript is now being handed over to our production team.

Kind regards, 

on behalf of

Dr. Noorsuzana Mohd Shariff 

Academic Editor

PLOS ONE